# De novo reconstruction of human adipose transcriptome reveals conserved lncRNAs as regulators of brown adipogenesis

Chunming Ding[1], Yen Ching Lim[1,2], Sook Yoong Chia[2], Arcinas Camille Esther Walet[2], Shaohai Xu [3], Kinyui Alice Lo[4], Yanling Zhao[1], Dewen Zhu[1], Zhihui Shan[1], Qingfeng Chen[4], Melvin Khee-Shing Leow[2,5,6,7], Dan Xu[1,2] & Lei Sun[2,4]

Obesity has emerged as an alarming health crisis due to its association with metabolic risk factors such as diabetes, dyslipidemia, and hypertension. Recent work has demonstrated the multifaceted roles of lncRNAs in regulating mouse adipose development, but their implication in human adipocytes remains largely unknown. Here we present a catalog of 3149 adipose active lncRNAs, of which 909 are specifically detected in brown adipose tissue (BAT) by performing deep RNA-seq on adult subcutaneous, omental white adipose tissue and fetal BATs. A total of 169 conserved human lncRNAs show positive correlation with their nearby mRNAs, and knockdown assay supports a role of lncRNAs in regulating their nearby mRNAs. The knockdown of one of those, *lnc-dPrdm16*, impairs brown adipocyte differentiation in vitro and a significant reduction of BAT-selective markers in in vivo. Together, our work provides a comprehensive human adipose catalog built from diverse fat depots and establishes a roadmap to facilitate the discovery of functional lncRNAs in adipocyte development.

[1] Key Laboratory of Laboratory Medicine, School of Laboratory Medicine and Life Science, Wenzhou Medical University, Wenzhou, Zhejiang 325035, China. [2] Cardiovascular and Metabolic Disorders Program, Duke-NUS Medical School, 8 College Road, Singapore 169857, Singapore. [3] Division of Bioengineering, Nanyang Technological University, 70 Nanyang Drive, Singapore 637457, Singapore. [4] Institute of Molecular and Cell Biology, 61 Biopolis Drive, Proteos, Singapore 138673, Singapore. [5] Clinical Nutrition Research Centre (CNRC), Singapore Institute for Clinical Sciences (SICS), Agency for Science, Technology and Research (A*STAR) and National University Health System (NUHS), Singapore 117599, Singapore. [6] National University Health System (NUHS), Singapore 119074, Singapore. [7] Department of Endocrinology, Tan Tock Seng Hospital, Singapore 308433, Singapore. These authors contribute equally: Chunming Ding, Yen Ching Lim. Correspondence and requests for materials should be addressed to C.D. (email: cmding@gmail.com) or to D.X. (email: dxuwzmu@gmail.com) or to L.S. (email: sun.lei@duke-nus.edu.sg)

The upsurge of modern obesity rate has raised a global public health alarm due to its increased risks for type 2 diabetes, cardiovascular diseases, stroke, hyperglycemia, dyslipidemia, hypertension, and cancers[1–3]. Elucidating molecular mechanisms underlying fat biology and regulating energy homeostasis play a critical role in combating obesity. Mammalian adipose tissue was historically classified as white adipose tissues (WATs) and brown adipose tissues (BAT). While WAT functions mainly to store excess energy as triglycerides, BAT is specialized for energy expenditure through the uncoupling of oxidative phosphorylation from ATP production by Ucp1[4,5]. Recent work uncovered clusters of Ucp1-expressing adipocytes, known as beige adipocytes, residing among subcutaneous WAT depots but not in visceral depots. Beige adipocytes show high phenotypic plasticity as they take on WAT morphology under basal state but show BAT-like morphology and thermogenic characteristics upon stimulation by cold exposure and agonists for β-adrenergic receptor or proliferator-activated receptor-γ (Ppar-γ)[6–8].

While BAT persists in rodents throughout lifetime, interscapular BAT in human exists only during infancy stage and degenerates as we grow. Thus, for decades, BAT was considered to be physiologically irrelevant and metabolically inconsequential to adult human[9,10]. However, recent positron-emission tomography/computed tomography scans studies[11–13] have revealed substantial depots of BAT in cervical, supraclavicular, and paravertebral regions[11–16], whose activity shows negative correlation with body mass index[11] and positive correlation with resting metabolism[16]. Hence, the re-discovery of BAT in adults has spurred immerse interest in targeting BAT as a non-invasive anti-obesity therapy. While infant BAT resembles rodent classical BAT[17], there is much debate over the cell type identities of human adult BAT. Previous studies have supported two different views: (i) coexistence of classical BAT and beige fats[18] in the depot; and (ii) analogous to rodent beige fat due to molecular resemblance[19,20]. Despite many publications on this topic, a consensus view has yet to emerge.

Emerging evidence derived from high-throughput sequencing has revealed a large number of non-coding transcripts[21]. Long non-coding RNAs (lncRNAs) are arbitrarily defined as transcripts longer than 200 bases with low protein-coding potential[22]. Despite the lack of protein-coding capacity, mounting evidence has pointed toward an epigenetic regulatory role of lncRNAs in diverse biological processes[22–24]. Earlier studies from our and other groups have revealed a set of lncRNAs essential for proper mouse WAT adipogenesis[5] and identified several BAT-selectively expressed lncRNAs such as lncBATE1, lncBATE10, and Blnc1 that are critical for BAT program expression[24–26]. Despite this progress, more efforts are needed to understand the function of lncRNAs in human adipocytes because most lncRNAs are poorly conserved between mouse and human.

A few studies are revealing regulatory roles of lncRNAs in human adipocytes. HOTAIR was reported to be expressed in gluteal adipose but almost absent in abdominal adipocytes. Overexpression of HOTAIR in abdominal adipocytes led to increased percentage of differentiated cells and enhanced expression of adipogenic markers such as PPARγ[27]. Another recent work using human mesenchymal stem cells demonstrated the depletion of ADINR, a lncRNA transcribed divergently from Cebpα, resulted in impaired adipogenesis[28]. Despite these progress, our understanding of lncRNA in human adipocytes has been hindered by the lack of a comprehensive lncRNA catalog, particularly in BAT due to the difficulty in sample collection. Databases such as GENCODE or University of California at Santa Cruz (UCSC) represent some of the best reputable lncRNA depositories, but the current annotation (~15.9k, Gencode version 24) undermines the lncRNA population due to intrinsic low abundance and high tissue specificity[29]. While the former problem can be overcome by performing deep sequencing, the latter poses a greater challenge since each cell type expresses its unique set of lncRNAs in a cell type-specific manner and the detectable lncRNA populations can be significantly different even for closely related cell types[30,31]. Thus, before the regulatory function of lncRNAs in human adipose and their therapeutic potential for obesity can be fully evaluated, it is needed to build a comprehensive human adipose lncRNA catalog.

To address these challenges, we perform deep RNA-sequencing (RNA-seq; ~682 million reads in total) on human fetal BAT, adult omental WAT (oWAT), and subcutaneous WAT (sWAT) to de novo reconstruct human adipose tissue transcriptomes. We uncover a total of 3149 lncRNAs, including 318 lncRNAs syntenically conserved between human and mouse. From this list, we identify a functional important lncRNA, lnc-dPrdm16, that is required for brown adipocyte differentiation. Our work provides a valuable resource of lncRNAs in human adipocytes and build a roadmap to facilitate the discovery of functional lncRNAs for adipocyte biology.

## Results

**Generation and characterization of human adipose lncRNA.** High-throughput sequencing coupled with computational pipelines have driven tremendous progress in the discovery of novel lncRNAs[24,32,33]. Nonetheless, the current lncRNA annotation is far from complete owing to its tissue-specific expression property. To better identify and characterize lncRNAs expressed in human adipose in vivo, we set out to de novo reconstruct the transcriptome by profiling human fetal BAT, oWAT, and sWAT (Fig. 1a, Supplementary Fig. 1A, Supplementary Table 1). We performed deep strand-specific, 100 bp paired-end sequencing on poly(A)-selected RNA and generated ~682.4 million reads. Reads were first mapped to Hg19 using Tophat[34] and subsequently input to Cufflinks[34] for transcriptome assembly and gene quantification. The precision of our RNA-seq data and de novo assembly were confirmed by examining the gene expression levels of pan, white, and brown fat markers (Fig. 1b) and predicted gene structures for Ucp1 and Leptin (Supplementary Fig. 1B).

We applied a stringent filter to focus only on long (>200 bp) transcripts that do not overlap with mRNA exons on the same strand (against UCSC, refSeq, and Ensembl databases) and show no evidence of protein-coding capacity (Fig. 1c). To prevent artifacts introduced by single-exonic fragments with low expression, only multi-exon transcripts were considered. By implementing this strategy, we identified 3149 lncRNAs, which exhibit similar characteristics as previously reported[24,29,32,33,35] such as lower gene expression (Fig. 1d), lower isoform number (Supplementary Fig. 1C), lower exon count (Supplementary Fig. 1D), shorter transcripts (Supplementary Fig. 1E), and shorter open reading frame (Supplementary Fig. 1F) than mRNAs. 1587 out of 3149 (50.40%) lncRNAs (Fig. 1e) while 1631 out of 14 383 (11.34%) mRNAs (Supplementary Fig. 1G) are detectable in only one of the three adipose subtypes, highlighting the tissue-specific expression nature of lncRNAs.

Given the exponential progress of lncRNA annotations in well-established database such as Gencode, we proceeded to assess the contribution of our de novo-reconstructed catalog to the existing knowledge base. Referencing against Gencode v24 human lncRNA annotation, we identified 2129 previously unannotated lncRNAs, accounting for more than two-thirds of newly built catalog. Analysis of the novel transcripts against the annotated counterparts revealed that novel lncRNAs have significantly fewer exons (two-sided Mann–Whitney U-test $p < 2.2 \times 10^{-16}$, Supplementary Fig. 1H) and isoforms (two-sided Mann–Whitney U-test

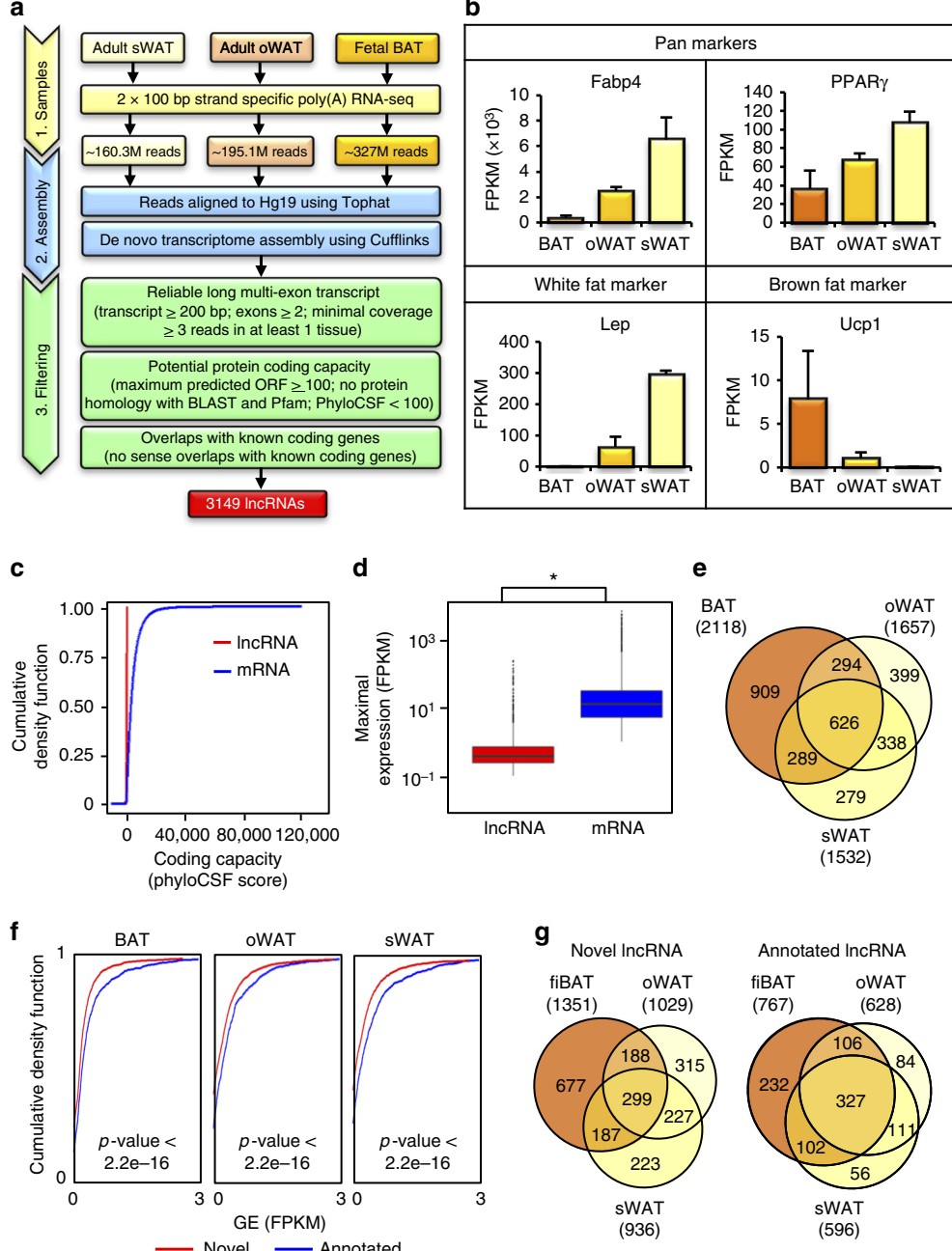

**Fig. 1** Generation and characterization of human adipose lncRNA. **a** Bioinformatics pipeline for human adipose lncRNA discovery pipeline. See "Material and methods" for details. **b** RNA-seq gene expression for pan markers (Fabp4 and Pparγ), white fat-specific marker (Leptin), and brown fat-specific marker (Ucp1). Error bars represent mean ± SEM. $n = 3/4$. **c** Coding potential of lncRNAs and mRNAs represented by PhyloCSF score. **d** Boxplots of the maximal expression distributions for adipose-expressed lncRNA (FPKM > 0.1) and mRNA (FPKM >1). Mann–Whitney $U$-test $p$-value < 2.2e-16. * denotes p-value < 0.05. **e** Overlaps of detectable lncRNAs between fiBAT, sWAT, and oWAT (FPKM > 0.1 in at least one of the three samples). **f** Cumulative density function of gene expressions for novel and annotated lncRNAs based on their abundance in BAT (left panel), oWAT (center panel), and sWAT (right panel). Novel lncRNAs exhibit significant lower expression than annotated ones in all three tissues. **g** Overlaps of detectable lncRNAs between fiBAT, oWAT, and sWAT (FPKM > 0.1 in at least one of the three samples). LncRNAs are classified by novel (left panel) and annotated (right panel) status

$p = 1.909e-15$, Supplementary Fig. 1I), shorter (Supplementary Fig. 1J) and lower expressed (Fig. 1F, Supplementary Fig. 1K) than annotated ones. Importantly, the proportion of tissue-unique lncRNAs were higher in novel (57.4%) than annotated category (36.5%; Fig. 1g, Supplementary Fig. 1L), confirming that current existing lncRNA database tends to miss tissue-specific lncRNAs. Taken together, we have provided a comprehensive catalog of human adipose lncRNAs, which are mostly unannotated.

**Tissue specificity of the human adipose lncRNAs.** To further investigate the tissue specificity of these lncRNAs, we computed their expression levels across a compendium of 19 non-adipose human tissues obtained from the Human BodyMap[33]. We assigned tissue specificity scores for each gene by calculating its fractional expression in each tissue against the summed expressions in all 22 tissues (Fig. 2a). Consistent with prior human and mouse studies[24,29,32,33,35], significant higher specificity scores

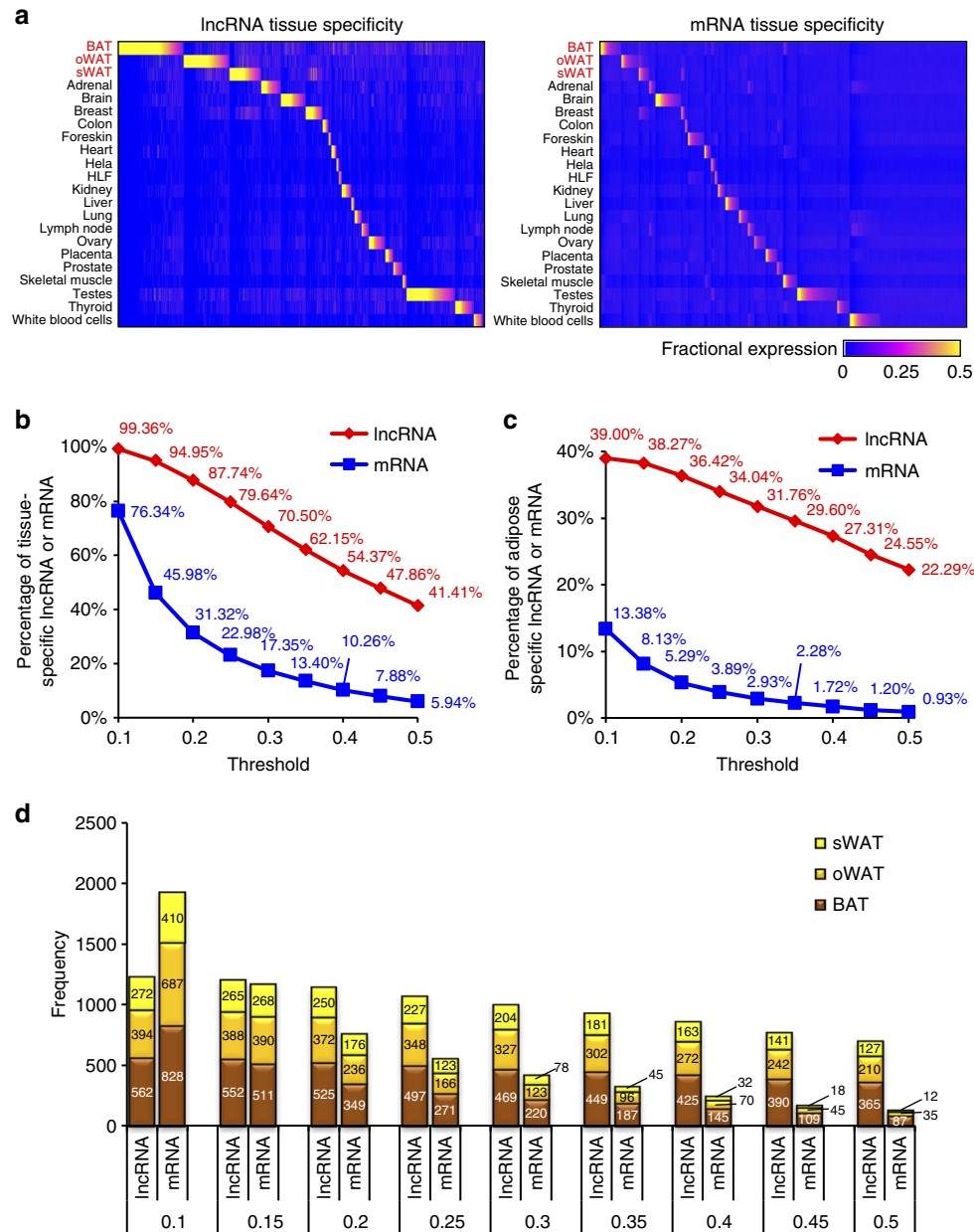

**Fig. 2** Tissue specificity of human adipose lncRNAs. **a** Relative abundance of 14 383 adipose-expressed mRNAs (rows, left panel) and 3149 lncRNAs (rows, right panel) across 22 examined tissues (columns). Color density represents the fractional expression of each lncRNA in each tissue relative to all examined tissues. **b** Percentage of tissue-specific lncRNAs and mRNAs under varying thresholds of fractional expression (step size 0.05). Tissue-specific transcripts in any of the 22 tissues were considered in this analysis. **c** Percentage of adipose-specific (BAT, oWAT, or sWAT) lncRNAs and mRNAs under varying thresholds of fractional expression (step size 0.05). Tissue-specific transcripts in any of the three adipose tissues were considered in the analysis. **d** Number of adipose-specific lncRNA and mRNA at varying thresholds. Tissue-specific transcripts for any of the three adipose tissues were considered in the analysis

were detected in lncRNAs (two-sided Mann–Whitney U-test $p < 2.2 \times 10^{-16}$, Supplementary Fig. 2A) than mRNAs. We imposed a maximal fractional expression (across 22 tissues) of at least 0.1 as the threshold to select for tissue-specific lncRNAs and found that 3129 (99.3%) lncRNAs, in contrast to 76.3% mRNA, passed the threshold, thus considered as tissue-specific (Supplementary Fig. 2B).

To test the robustness of tissue-specific nature of lncRNAs, we approached the problem from two perspectives: (i) we varied the maximal fractional expression threshold from 0.1 to 0.5 with uniform step size of 0.05; and (ii) standardizing gene expression cutoffs for lncRNAs and mRNAs. Using the first method, lncRNAs were found to perform consistently better than mRNAs

in tissue specificity, in both scenarios when considering tissue-specific transcripts in all 22 tissues (Fig. 2b) and the subset of adipose tissue-specific transcripts (Fig. 2c). The differences between the two classes of RNAs were more pronounced at higher thresholds (Fig. 2b–d) and when only considering adipose-specific lncRNAs (Fig. 2c, d). For example, when a maximal fractional expression threshold of 0.5 was imposed, we found that tissue-specific lncRNAs (41.41%) were approximately 6 times more than mRNAs (5.94%; Fig. 2b), and when only adipose-specific subset was considered, lncRNAs (22.29%) were about 23 times more than mRNAs (0.93%; Fig. 2c).

Because lncRNAs were lower expressed than mRNAs[24,29,32,33,35], a higher cutoff (fragments per kilobase of exon per million

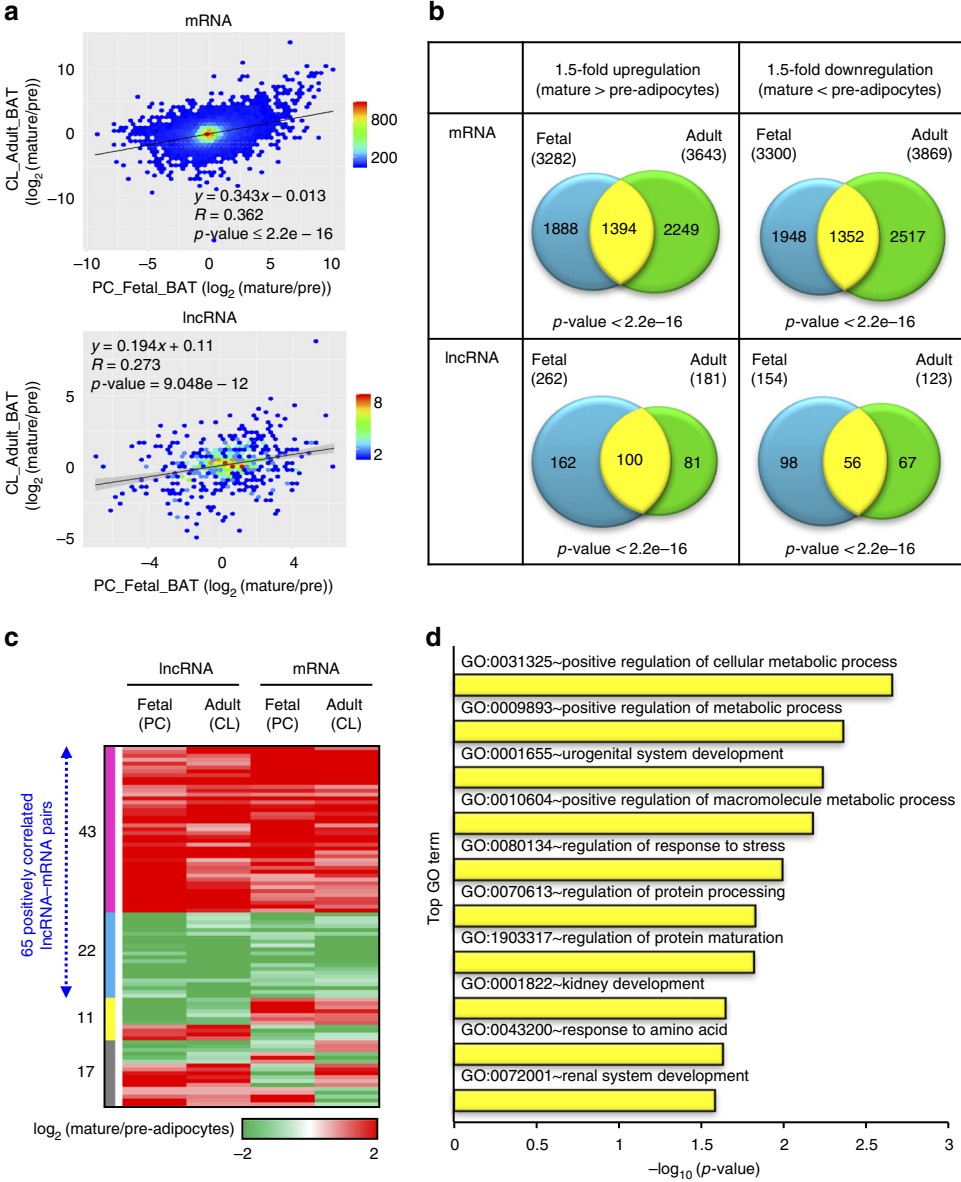

**Fig. 3** Comparison of global expression changes between fetal and adult brown adipogenesis. **a** Hexbin plot of global gene expression changes for 14 487 mRNAs (top panel) and 586 lncRNAs (bottom panel) with detectable expressions during both fetal and adult BAT adipogenesis. Density of the data points is represented by color scales. PC primary cells, CL cell lines. **b** Venn diagram comparing fetal and adult BAT's transcripts that are regulated (1.5-fold) during adipogenesis. mRNAs (top) and lncRNAs (bottom). **c** Heatmap depicting 93 lncRNA–mRNA pairs having at least 1.5-fold change between pre-adipocytes and mature adipocytes for both primary cells and cell lines. Pairs are grouped by the lncRNA–mRNA correlation status. Blue and magenta bars represent the lncRNA–mRNA pairs with coherent down- and upregulations during brown adipogenesis. Yellow bar represents the pairs for which the lncRNAs show anti-correlation with mRNA while gray bar represents the pairs, which show incoherent correlation. **d** Top 10 enriched gene ontology processes of the mRNAs in the upregulated lncRNA–mRNA pairs during both fetal and adult BAT adipogenesis (magenta category from **c**)

fragments mapped (FPKM) ≥ 1) was used to select the subset of adipose-expressed mRNAs in above analysis. To test if the tissue-specific characteristic of lncRNA is an artifact introduced by imposing unstandardized threshold expression, we analyzed the set of lncRNAs (Supplementary Fig. 2C) using a higher threshold (FPKM ≥ 1) and mRNAs (Supplementary Fig. 2D) at a lower threshold (FPKM ≥ 0.1). At both low (Supplementary Fig. 2E) and high (Supplementary Fig. 2F) expression thresholds, the proportion of tissue-specific lncRNAs is consistently higher than that of mRNAs.

**Dynamic changes of lncRNA expression during BAT adipogenesis.** We used a previously established protocol (Methods)[36]

to differentiate isolated pre-adipocytes from non-viable human fetal BAT and from adult subcutaneous WAT into mature adipocytes (Supplementary Table. 1). We performed RNA-seq on pre- and mature adipocytes to examine the transcriptome dynamic of lncRNA and mRNA during adipogenesis. Performance of global gene ontology analyses (gene set enrichment analysis; GSEA[37]) on regulated genes during brown adipocyte differentiation revealed significant enrichment in adipogenesis and lipid metabolism terms (Supplementary Fig. 3A), thereby confirming the biological validity of the adipogenesis model.

To examine the transcriptomic profile differences between fetal and adult BAT, we integrated our data with publicly available brown adipogenic RNA-seq profiles derived from

human adult clonal cell lines (ArrayExpress, accession number E-MTAB-2602)[19]. The gene expression changes of lncRNAs during adipogenesis were more dynamic than those of mRNAs in all examined cell types (Supplementary Fig. 3B, C). To address concerns regarding possible confounding effects introduced by inter-individual variation and analyzing datasets derived from primary cells and clonal cell lines (Supplementary Fig. 3D, E), we compared the transcriptomes of human primary white adipocytes with clonal white adipocyte cell lines (ArrayExpress, accession number E-MTAB-2602)[19]. Scatterplot of expression changes of protein-coding genes between WAT primary cells and cell lines displayed a significant correlation ($R = 0.519$, $p$-value < 2.2e-16, Supplementary Fig. 3F). This indicates a decent level of similarity between the datasets from primary cells and cell lines obtained from different individuals, thus the comparison between the fetal primary brown adipocytes and the adult clonal cell line is reasonable. Correlation of gene expression changes between adult and fetal BAT adipogenesis was positive and significant for both mRNA ($R = 0.362$, $p$-value < 2.2e-16) and lncRNA ($R = 0.273$, $p$-value = 9.048e-12; Fig. 3a). We overlapped the genes that were regulated by at least 1.5-fold in two cell types and found that the overlapping was significant for all four comparisons ($p < 2.2e-16$, hypergeometric test, Fig. 3b), suggesting that the transcriptome

changes during fetal and adult brown adipogenesis, despite from different lineage origin, largely resemble each other.

We next examined the correlation between lncRNAs and their nearby mRNAs that are up- or downregulated (>1.5-fold) during both fetal and adult BAT adipogenesis. In all, 65 out of 93 lncRNA–mRNA pairs exhibit positive correlation, including 43 upregulated and 22 downregulated pairs (Fig. 3c), suggesting that there might be in *cis* regulatory interactions between lncRNAs and their nearby mRNAs. Notably, these upregulated protein-coding mRNAs were highly associated with "positive regulation of cellular metabolic process" (Fig. 3d).

**LncRNAs are dynamically regulated in adult BAT during cold exposure**. To better understand the dynamics of lncRNA expression change in human BAT during cold activation, we analyzed the RNA-seq data of adult BAT derived from an individual at thermoneutrality and after 5-h cold exposure, while subcutaneous abdominal WAT was used as a control (ArrayExpress, accession number E-MTAB-4031)[38]. Analysis of expression profiles showed that while lncRNAs are more dynamic than mRNAs, there was an obvious dominance of downregulated lncRNAs and mRNAs in both adipose tissues, particularly BAT

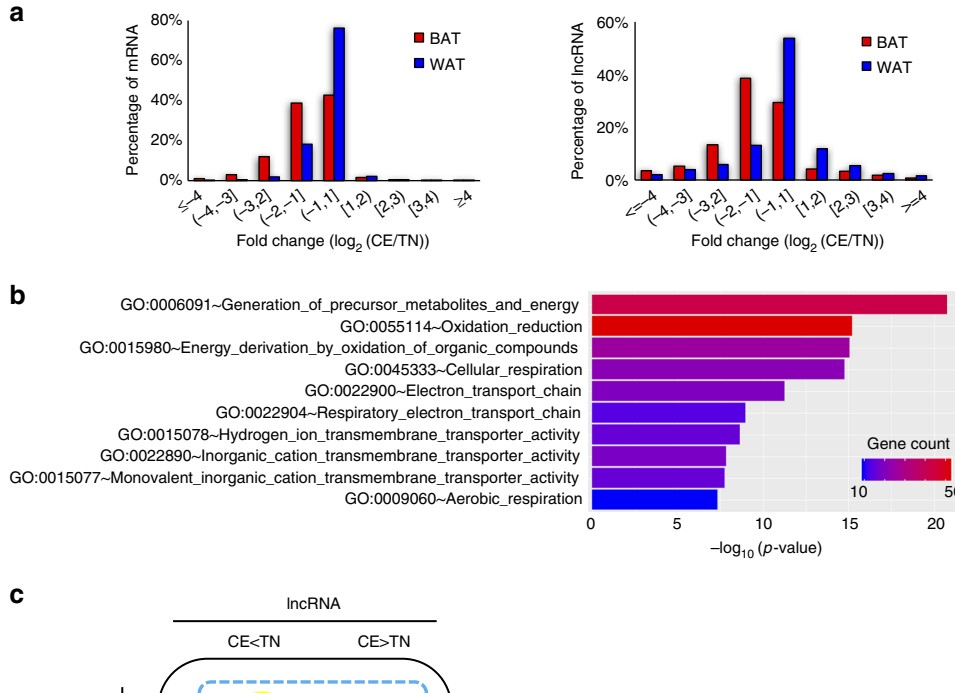

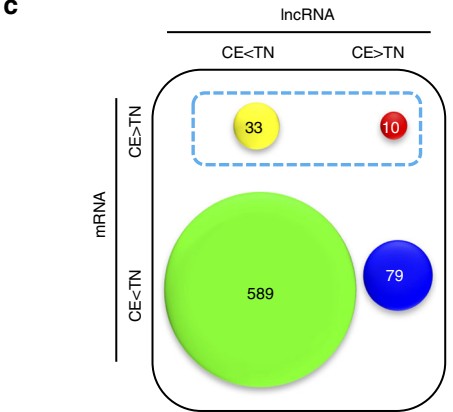

**Fig. 4** Comparison of global expression changes upon cold exposure (relative to thermoneutral conditions) in human BAT and WAT tissues. **a** Distribution of gene expression changes upon cold exposure for lncRNA (left panel) and mRNA (right panel). **b** Top 10 enriched gene ontology processes of the upregulated mRNAs in BAT upon cold exposure. **c** 2 by 2 matrix depicting the gene expression changes in adult BAT upon cold exposure for lncRNA–mRNA pairs. Size of each circle is proportional to the number of lncRNA–mRNA pairs. CE cold exposure, TN thermoneutral

(Fig. 4a). Global gene ontology analyses indicated that upregulated genes in BAT upon cold exposure were significantly enriched for processes such as cellular respiration, oxidation reduction, and electron transport chain (Fig. 4b, Supplementary Fig. 4A), while evidence for metabolism changes was absent in WAT (Supplementary Fig. 4B, 4C). We next assessed the expression correlation between lncRNAs and their nearby protein-coding genes by extracting lncRNA–mRNA pairs with more than 1.5-fold expression changes. We observed as many as 599 out of 711 (84.2%) of the lncRNA–mRNA pairs being positively correlated (Fig. 4c), suggesting that the paired lncRNAs and mRNAs may regulate the transcription of each other positively in *cis* or they may share common regulatory elements.

**Conserved lncRNAs are expressed more broadly in multiple tissues than non-conserved ones.** Although traditional judgment of evolutionary conservation by primary sequence has been successful for protein-coding genes, this approach is less effective in lncRNAs, which display low sequence homology between species[39,40]. Sole reliance on sequence conservation could lead to an

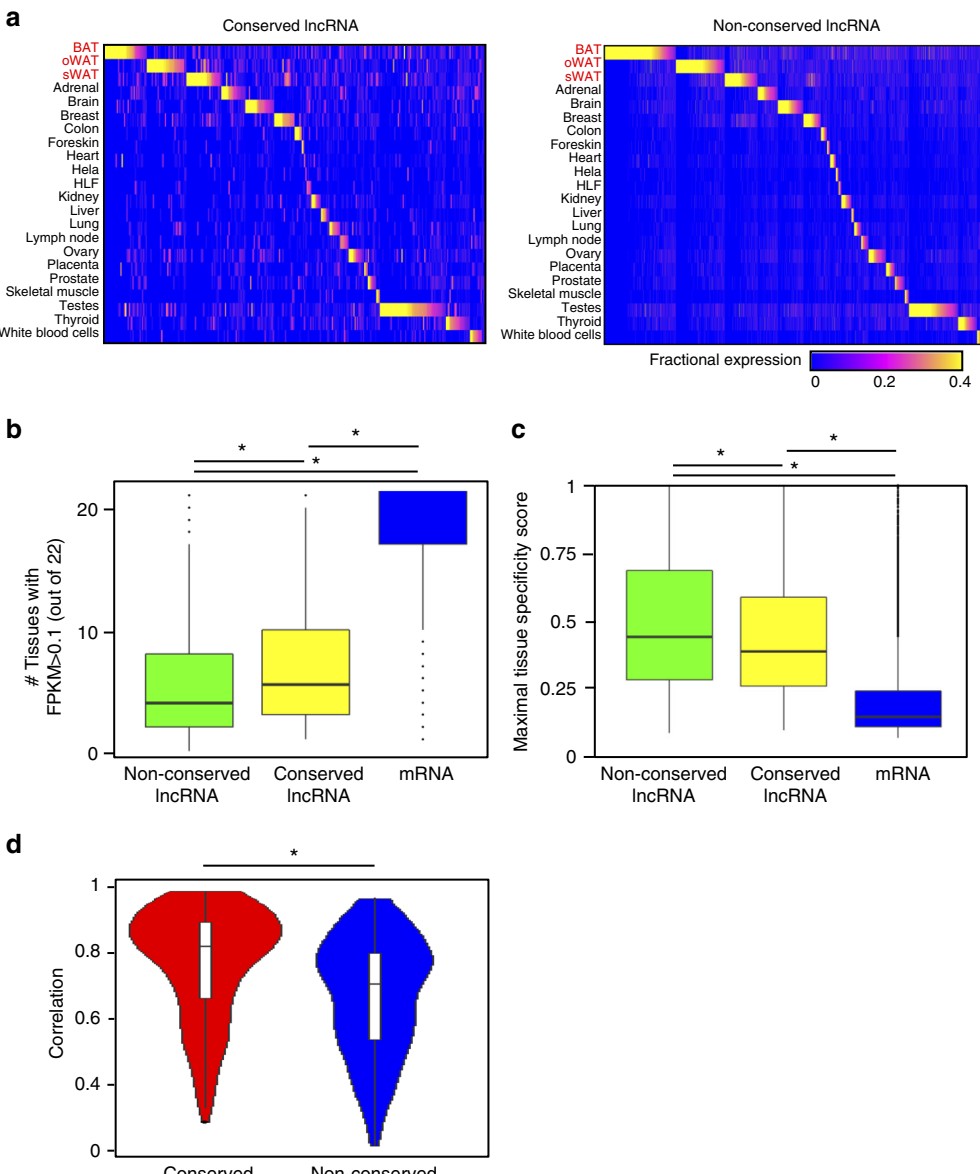

**Fig. 5** Conserved lncRNAs are more broadly expression in more tissues types. **a** Tissue specificity of 318 conserved (left panel) and 2831 non-conserved (right panel) lncRNAs across 22 examined human tissues. **b** Boxplots comparing the counts of tissues in which non-conserved lncRNAs, conserved lncRNAs, and mRNAs can be detected (FPKM > 0.1). Tissue counts refer to the number of tissues (out of 22) that show expression level FPKM > 0.1 for each category of transcripts. Non-conserved lncRNAs have the lowest tissue counts, followed by conserved lncRNAs and mRNAs. Non-conserved lncRNAs vs. conserved lncRNAs: *p*-value = 0.0001587; non-conserved lncRNAs vs. mRNAs: *p*-value < 2.2e-16; conserved lncRNAs vs. mRNAs: *p*-value < 2.2e-16. Mann–Whitney *U*-test. **c** Boxplots comparing maximal tissue specificity scores among non-conserved lncRNAs, conserved lncRNAs, and mRNAs. Maximal tissue specificity score is defined by the maximal fractional expression of the examined lncRNAs among all 22 tissues. Non-conserved lncRNAs show the highest tissue specificity, followed by conserved lncRNAs and mRNAs. Non-conserved vs. conserved, *p*-value = 0.003371; non-conserved vs. mRNA, *p*-value < 2.2e-16; conserved vs. mRNA, *p*-value < 2.2e-16. Mann–Whitney *U*-test. **d** Violin plots representing the distribution of all possible pairwise correlations among 22 tissues, using either conserved or non-conserved lncRNAs. Correlations between tissues generated using conserved lncRNAs are significantly higher than using non-conserved lncRNAs. Mann–Whitney *U*-test, *p*-value = 3.362e-11. * denotes significant p-value < 0.05

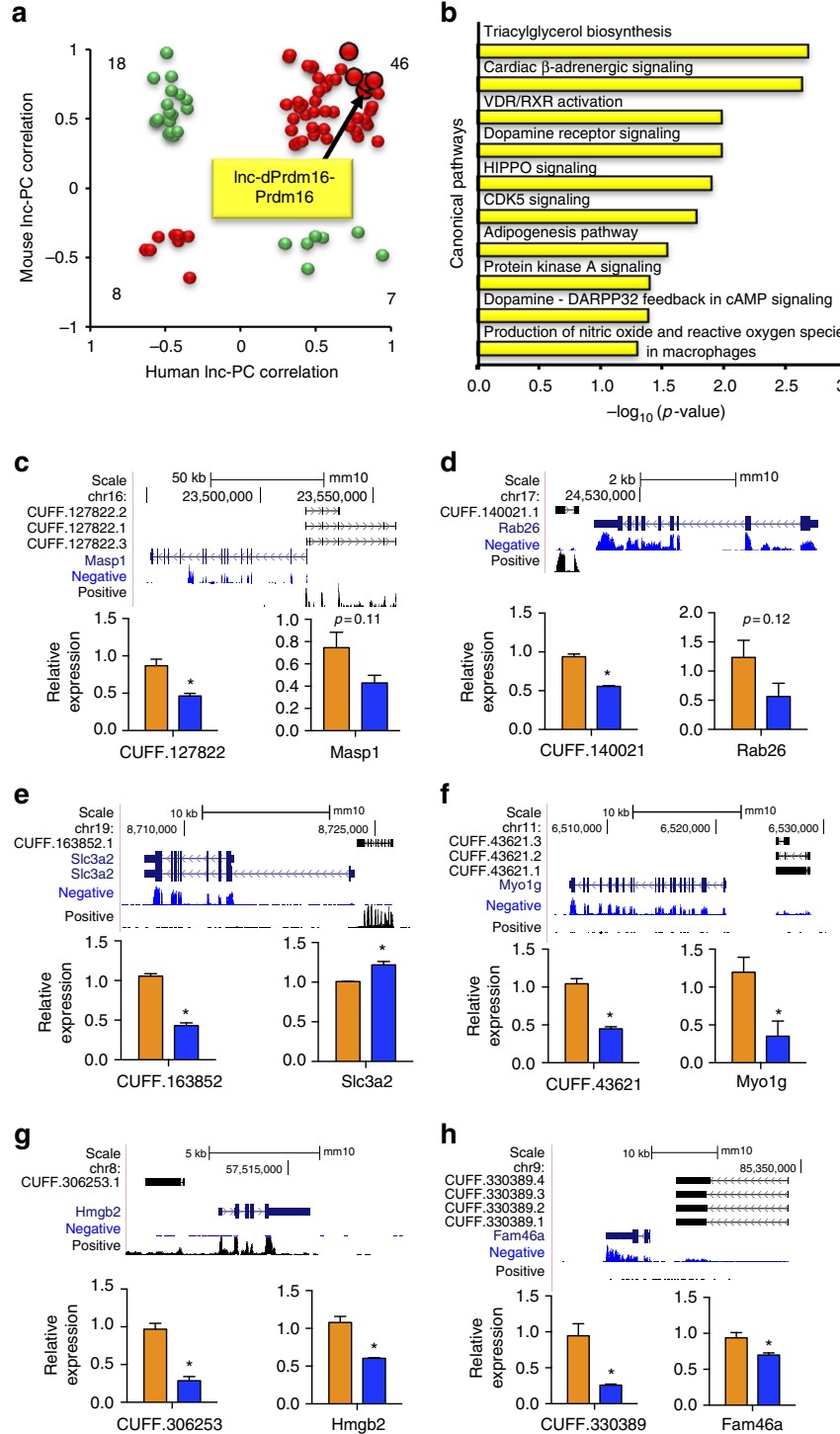

**Fig. 6** Regulatory interactions between lncRNAs and their nearby mRNAs. **a** Scatterplot of lncRNA–mRNA correlations in mouse against those in human. Only correlation values of at least 0.3 in both species are included in the plot. In all, 54 out of 79 pairs (red dots) show conserved correlations between mouse and human. Binomial test, *p*-value = 0.001466. **b** Top enriched gene ontology terms of the protein-coding genes extracted from the upper right quadrant of **a**. **c–h** siRNA knockdown of six selected lncRNAs in brown pre-adipocytes followed by 4 days of differentiation. Real-time PCR was performed to detect the knockdown efficiency and the expression change of their nearby mRNAs. Gene expression was normalized to *RPL23* as the housekeeping gene. Error bars represent mean ± SEM, *n* = 3, **p* < 0.05, Student's *T*-test

underestimation of the conserved lncRNA population. Here we used both sequence similarity and genomic synteny (Supplementary Fig. 5A) to compare newly constructed human vs. previously constructed mouse catalogs[24] and identify conserved adipose-expressed lncRNAs that satisfy either criteria. Based on these methods, only 318 conserved lncRNAs (10.1% of human

lncRNAs and 19.5% of mouse lncRNAs) were identified, suggesting that majority of the lncRNAs are species-specific. Between conserved and non-conserved lncRNAs, there was no apparent difference in exon (Supplementary Fig. 5B) and isoform (Supplementary Fig. 5C) distributions, gene expression (Supplementary Fig. 5D), variation (Supplementary Fig. 5E), and correlation

with neighboring protein-coding genes (Supplementary Fig. 5F). To compare the tissue specificity of conserved and non-conserved lncRNAs, we examined the fractional expressions of each lncRNA across the tissue compendium and found that conserved lncRNAs tend to be more broadly expressed in multiple tissues (Fig. 5a, b) and hence have lower tissue specificity scores (Fig. 5c). Intrigued by this observation, we generated all possible pairwise tissue combinations from the 22-tissue panels, and calculated Pearson correlation values for each tissue pair based on lncRNA expression. By collating correlation values from all possible tissue pairs, we plotted the distribution of Pearson correlations for both conserved and non-conserved lncRNA categories. Congruent with earlier results from Fig. 5a–c, pairwise correlations of conserved lncRNAs are significantly higher ($p$-value = 3.362e-11) than those of non-conserved lncRNAs (Fig. 5d). These results demonstrated that conserved lncRNAs tend to be more ubiquitously expressed across different cell types and are likely to be essential for more basic cellular functions[41].

**Functional prediction of a conserved lncRNA, lnc-dPrdm16.** Global analysis of correlations between lncRNAs and their flanking mRNA in both mouse and human demonstrates a tendency of positive correlation (Supplementary Fig. 6A, B). To test whether the correlative relationship between lncRNA–mRNA pairs persists during evolution, we extracted 79 conserved pairs with $R$ value ≥ 0.3 in both species and compared their correlations between mouse and human. Remarkably, 54 out of 79 pairs (65.8%, two-sided binomial $p$-value = 0.00095) showed conserved

correlation, of which 46 of 54 pairs (85.2%) exhibited positive correlation (Fig. 6a) and were significantly enriched for adipose-related canonical pathways such as triacylglycerol biosynthesis and adipogenesis (Fig. 6b).

To test whether the positive correlation between lncRNA–mRNA pairs reflects a regulatory relationship or just a share of common regulatory elements, we conducted small interfering RNA (siRNA) knockdown for six lncRNAs in brown adipocyte culture. In as many as five out of six knockdown lncRNAs, the nearby mRNA was also downregulated (Fig. 6c–h), strongly supporting that lncRNAs may have positive influence on the transcription of their nearby mRNAs. However, the influence of these lncRNAs may or may not rely on their sequences per se. It was reported that the general process of these transcripts such as splicing may mediate the cross talk between lncRNA–mRNA pairs[42], which warrants further study. It is also possible that some lncRNAs do not affect their nearby mRNAs directly but through secondary effects from altered cellular status.

From these conserved lncRNA–mRNA pairs, we identified an interesting lncRNA, which is located divergently from *Prdm16*, a master regulator for brown adipocyte biology (Fig. 7a) (hereafter referred as *lnc-dPrdm16*) and corresponded to a known annotated lncRNA, LINC00982. In addition to their location proximity, *lnc-dPrdm16* also exhibits a high correlation in human ($R = 0.829$; Fig. 7b, left panel) and mouse ($R = 0.702$; Fig. 7b, right panel) with *Prdm16*. We examined its subcellular localization in BAT and found this lncRNA, unlike most other lncRNAs, is mainly localized in cytosol (Supplementary Fig. 7).

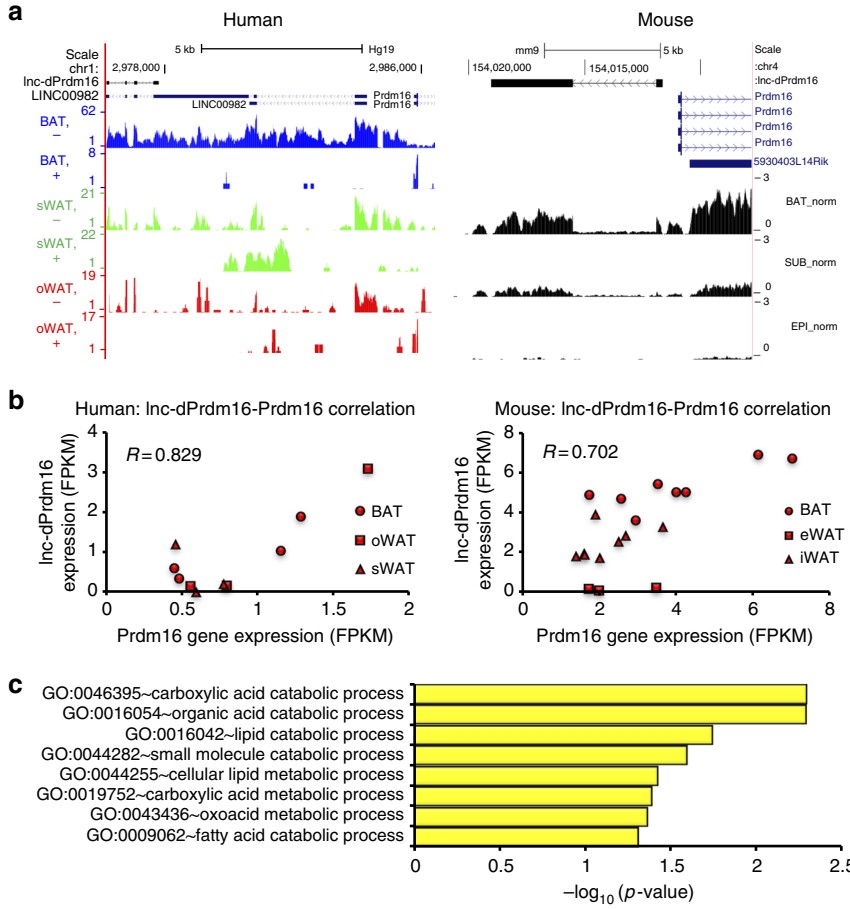

**Fig. 7** *lnc-dPrdm16* is predicted as a regulator in lipid metabolism. **a** Locus map depicting the relative location of *lnc-dPrdm16* (corresponds to annotated LINC00982) to *Prdm16* in human (left panel) and mouse (right panel). **b** Data points of gene expression of *lnc-dPrdm16* and *Prdm16* across different human (left panel) and mouse samples (right panel). **c** Top enriched GO terms for protein-encoding genes that are positively co-expressed with lnc-dPrdm16

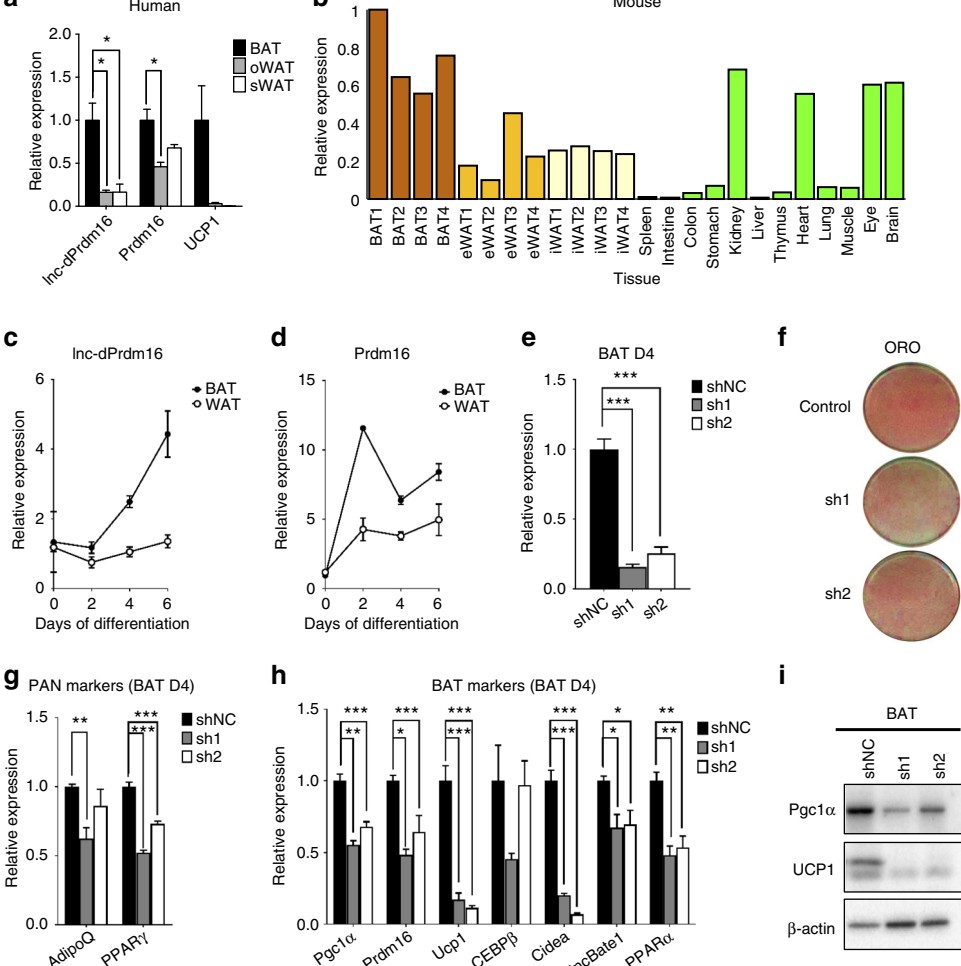

**Fig. 8** The expression of *lnc-dPrdm16* is needed for brown adipogenesis. **a** Real-time quantitative PCR analyses of *lnc-dPrdm16* expression in human BAT, oWAT, and sWAT. Error bars represent mean ± SEM, *n* = 2/3. **b** Real-time quantitative PCR of *lnc-dPrdm16* expression across 15 mouse tissues. **c** Real-time quantitative PCR analyses of *lnc-dPrdm16* during mouse BAT and WAT adipogenesis. Error bars represent mean ± SEM, *n* = 4. **d** Real-time quantitative PCR analyses of *Prdm16* during mouse BAT and WAT adipogenesis. **e** Knockdown efficiency of *lnc-dPrdm16* in cultured brown adipocytes, which were infected by retroviral shRNAs at precursor stages and then induced to differentiate for 4 days. Error bars represent mean ± SEM, *n* = 4. **f** Oil Red O staining at day 4 shows *lnc-dPrdm16* knockdown reduces lipid accumulation, **g** *lnc-dPrdm16* knockdown reduces PAN markers in BAT at day 4. Error bars represent mean ± SEM, *n* = 4. **h** *lnc-dPrdm16* knockdown reduces BAT markers in BAT at day 4. Error bars represent mean ± SEM, *n* = 4. **i** Western blot to examine the protein levels of *Pgc1α* and *Ucp1* on cultured day 4 brown adipocytes, which were infected by retroviral shRNAs at precursor stage. *$p \leq 0.05$, **$p \leq 0.01$, ***$p \leq 0.001$ (*T*-test)

Because functionally related genes in a common pathway are likely to be regulated in a similar fashion, we attempted to infer specific roles of *lnc-dPrdm16* by using known functions of highly co-expressed protein-coding genes. Among the top significantly enriched pathways of these positively correlated genes ($R \geq 0.7$) include "lipid catabolic process", "cellular lipid metabolic process", and "fatty acid catabolic process" (Fig. 7c), supporting the potential engagement of *lnc-dPrdm16* in adipocyte biology. Here we provide a roadmap to identify conserved lncRNAs with high potentiality for adipose function by integrating correlation conservation and functional prediction from co-expressed protein-coding genes.

**Lnc-dPrdm16 knockdown blocks adipogenesis in vitro**. Above analysis has led us to identify *lnc-dPrdm16* as a strong candidate in modulating adipose biology. We next validated the expression of *lnc-dPrdm16* in human adipose tissues by quantitative real-time PCR (qPCR). Not only is *lnc-dPrdm16* significantly higher

expressed in BAT than oWAT or sWAT, the level of difference is more prominent than *Prdm16* (Fig. 8a). Consistent with human data, expression of mouse ortholog of *lnc-dPrdm16* across a panel of 15 tissues also revealed an apparent enrichment in BAT than the other two fat tissues (Fig. 8b). We also monitored the expressions of *lnc-dPrdm16* (Fig. 8c) and *Prdm16* (Fig. 8d) during adipogenesis in mouse BAT and WAT cell cultures, and both were found to be upregulated in mature adipocytes.

To determine its biological function, we knocked down this lncRNA in primary brown adipocyte culture by retroviral small hairpin RNA (shRNA). Over 70% knockdown was achieved at day 5 (Fig. 8e), resulting in reduction in lipid accumulation (Fig. 8f), and downregulation of both general adipogenic markers (*AdipoQ* and *PPARγ*; Fig. 8g) and brown fat markers (*Pgc1α*, *Ucp1*, *Cebpβ*, *Cidea*, *lncBATE1*, *Prdm16*, and *PPARα*; Fig. 8h). We further confirmed a reduction in *Pgc1α* and *Ucp1* protein levels by immunoblotting (Fig. 8i, Supplementary Fig. 10). Taken together, we show that loss of *lnc-dPrdm16* function resulted in reduction of Prdm16 expression and obvious impediment of brown

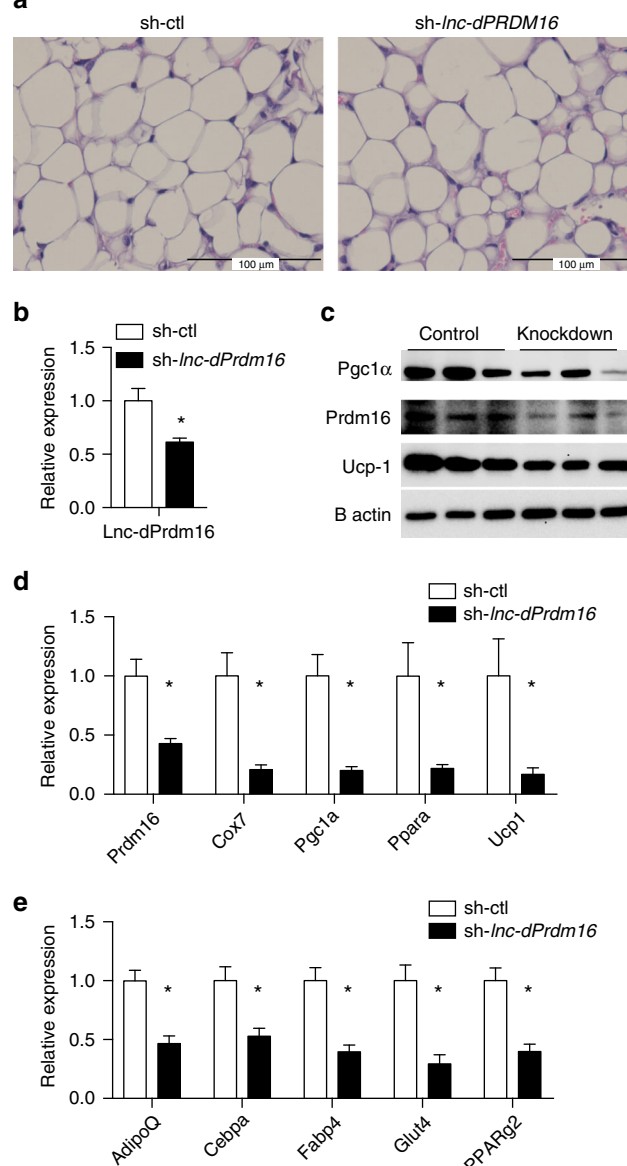

**Fig. 9** The expression of *lnc-dPrdm16* is required for iWAT browning. **a** Mice were infected by adenoviral sh-ctl and sh-*lnc-dPrdm16* for 7 days, and then exposed to 4 °C for 24 h. H&E staining was conducted to examine the cellular morphology in iWAT tissue. **b** Real-time PCR to examine the expression change of *lnc-dPrdm16*. sh-ctl, n = 8, sh-*lnc-dPrdm16*, n = 5. **c** Western blot to examine the protein level change of BAT-selective markers. **d**, **e** Real-time PCR to examine the BAT-selective marker expression (**d**) and pan-adipogenic markers (**e**). sh-ctl, n = 8, sh-*lnc-dPrdm16*, n = 5. Error bars represent mean ± SEM, *p < 0.05, Student's *T*-test

adipogenesis. To test whether *lnc-dPrdm16* is required for general adipogenesis, we also knocked down lnc-dPrdm16 in primary white adipocyte culture and also observed a block of differentiation during white adipogenesis (Supplementary Fig. 8).

**lnc-dPrdm16 is a regulator in BAT and during inguinal WAT browning in vivo**. To understand the function of *lnc-dPrdm16* in mature adipocytes in vivo, we constructed an adenovirus shRNA targeting this lncRNA and locally injected this virus as well as a control virus into mouse inguinal WAT (iWAT), which is mainly composed of mature adipocytes. One week after injection, these animals were exposed to 4 °C for 24 h to induce browning. We

did not observe significant difference in cellular morphology in Hematoxylin and eosin assay (Fig. 9a), but under fluorescent microscope, green fluorescent protein (GFP) expression in almost the entire isolated iWAT was readily detected (Supplementary Fig. 9A), indicating a high infection efficiency in vivo. We detected a significant reduction in *lnc-dPrdm16* (Fig. 9b, Supplementary Fig. 11), which was accompanied by decreased expression of BAT-selective and pan-adipogenic markers at both mRNA and protein levels (Fig. 9c–e). Therefore, lnc-dPrdm16 is required for maintaining a full mature adipocyte program in WAT and for BAT-marker induction during iWAT browning. We conducted similar experiments in BAT and observed that knockdown of *lnc-dPrdm16* impaired the expression of BAT-selective markers in BAT but did not affect the pan-adipogenic markers in BAT (Supplementary Fig. 9B-D). It is also worth noting that knockdown of *lnc-dPrdm16* resulted in decreased expression of *Prdm16* during iWAT browning (Fig. 9c, d) but not in interscapular BAT, which suggests that the regulatory interaction between lnc-dPrdm16 and Prdm16 is depot-specific, and *lnc-dPrdm16* can influence BAT-selective program in a Prdm16-dependent or -independent manner given different cellular contexts.

## Discussion

The global escalation of obesity rate has emerged as one of the most worrying health issues due to its accompanying higher risks for serious health disorders. Elucidating regulatory factors that control adipogenesis has thus presented as an attractive strategy to combat obesity. LncRNAs stood out as potential targets for novel therapies because of their diverse roles in regulating biological processes[43,44]. While we have seen significant progress in deciphering roles of lncRNAs in modulating fat biology in mouse[5,24], such knowledge cannot be easily extrapolated and verified on human studies due to limited availability of infant and adult BAT. Here we present the first comprehensive de novo-reconstructed lncRNA catalog using adult BWTs and WATs, which uncovered more than 2000 previously unannotated lncRNAs. This provides an additional and valuable resource for lncRNA study in human.

By integrating our human adipose-expressed lncRNAs with previously established mouse adipose catalogs[24], we identified 318 human lncRNAs (~10%), which showed either sequence homology or syntenic similarity with mouse lncRNAs. Interestingly, the co-expression correlation between these conserved lncRNAs and their nearby mRNAs tends to be preserved between species. We identified and functionally characterized *lnc-dPRM16*, a lncRNA located divergently from Prdm16[45]. Loss-of-function studies of *lnc-dPRM16* in mouse cultures not only downregulate Prdm16 expression but also markedly repress brown adipogenesis in cell culture (Fig. 8). Adenovirus-mediated knockdown of lnc-dPrdm16 in vivo preferentially affects BAT-selective markers but not pan-adipogenic markers in BAT (Supplementary Fig. 9); loss of lnc-dPrdm16 in iWAT resulted in reduced expression of pan-adipogenic markers and BAT-selective markers (Fig. 9). Moreover, loss of *lnc-dPrdm16* caused decreased expression of *Prdm16* in iWAT but not in BAT (Fig. 9, Supplementary S9). Together, our data clearly established *lnc-dPrdm16* as an essential non-coding gene in regulating adipogenesis and maintaining pan-adipocyte and/or BAT-selective program in mature adipocytes. However, how it can achieve its depot-specific influence and whether it can function independently from Prdm16 should be investigated in the future.

We estimated that ~10% of lncRNAs in human adipose have corresponding compartments in mouse, but this may underestimate the number of conserved lncRNAs. Because conserved synteny was assessed by relative locations of flanking

protein-coding genes found ±500 kb on the same strand, lncRNAs found on gene desert regions would be precluded. In fact, as many as ~47% of our human adipose lncRNAs fell within this category and thus eliminated from downstream analysis. Because our approach sets a stringent criterion to minimize false positive, we might have underestimated the true population of conserved lncRNAs. In light of this, it may be important to also explore conserved secondary and tertiary structures, which are critical for their regulatory functions[39].

## Methods

**Ethics statement**. Human fetal BAT was obtained from Advanced Bioscience Resources (Alameda, CA) from deceased donors as approved under exemption 4 in the HHS regulations (45 CFR Part 46). ABR follows established procedures for written informed parental consent. Zen-Bio conducted basic research in accordance with NIH guidelines and the Federal Provisions Pertaining to Research Use of Human Fetal Tissue by NIH Investigators. Zen-Bio's research related to human tissues is approved under its Institutional Review Board (IRB) through PearlIRB.

**Human adipose RNA samples and human primary adipocyte culture**. RNA samples of human BAT, oWAT, and sWAT were obtained from Zen-Bio Inc. (RNA-T10-CS., RNA-OM-T10, and RNA-T10−1).

White adipocyte precursors were obtained from Zen-Bio Inc. (SP-F-1). Brown adipocyte precursors were isolated from BAT as described previously[36]. Briefly, BAT was transported in fetal bovine serum (FBS) and antibiotic-containing Dulbecco's modified Eagle medium (DMEM) and then minced in a solution containing 1% collagenase, 1% phosphate-buffered saline (PBS), and 1% bovine serum albumin. Minced tissues were incubated for 45 min at 37 °C with gentle inversion. The solution was then strained through a cell strainer, supplemented with serum-containing medium, and centrifuged to collect a brown pre-adipocyte cell pellet.

Brown and white pre-adipocytes were cultured and differentiated according to previously described protocol[36]. Briefly, primary human pre-adipocytes isolated from adult subcutaneous and fetal interscapular tissue were plated in DMEM/Ham's F-12 medium (1:1, v/v) supplemented with HEPES (pH 7.4), FBS 10%, penicillin and streptomycin (1:100), and amphotericin B (1 μg/ml). Once cells reached confluence, they are induced to differentiate using differentiation medium (DMEM, FBS 10%, IMBX 0.25 mM, dexamethasone 0.5 mM, insulin 850 nM, indomethacin 100 μM, rosiglitazone 1 μM, penicillin and streptomycin (1:100), and fungizone/amphotericin B (1 μg/ml)). Cells were maintained in this media for 7 days after which the media was changed to maintenance medium (DMEM, FBS 10%, insulin 160 nM, penicillin and streptomycin (1:100), and fungizone/amphotericin B (1 μg/ml)). Cells were maintained in this manner with media changed every 2 days until their use in experiments at approximately 21 days post differentiation.

**Rodent primary adipocyte culture and differentiation**. Interscapular BWTs or iWATs from six to eight ~3-week-old mice were pooled together, minced, and digested in 0.2% collagenase, which were subsequently filtered by 40 μm cell strainer and centrifuged to collect stromal vascular fraction (SVF) cells at the bottom. SVF cells were cultured for downstream experiments.

Primary SVF cells were cultured in DMEM with 10% new-born calf serum until confluence. Cells were induced to differentiate for 2 days with DMEM containing 10% FBS (Invitrogen), 850 nM insulin (Sigma), 0.5 μM dexamethasone (Sigma), 250 μM 3-isobutyl-1-methylxanthine, phosphodiesterase inhibitor (IBMX, Sigma), 1 μM rosiglitazone (Cayman Chemical), and 1 nM T3 (Sigma). The induction medium was replaced with DMEM containing 10% FBS and 160 nM insulin for 2 days. Then cells were incubated in DMEM with 10% FBS.

**Plasmid and constructs**. All the plasmids used in this study were cloned using standard method. shRNAs targeting *lnc-dPrdm16* and a control shRNA were cloned into a retroviral vector (pMKO-GFP, Addgene 10676)

Neg control:
CAACAAGATGAAGAGCACCAACTCGAGTTGGTGCTCTTCATCTTGTTG
sh-*lnc-dPrdm16* 1:
GCAGCTTGATTACTTACAAGActcgagTCTTGTAAGTAATCAAGCTGC
sh-*lnc-dPrdm16* 2:
GGACTAACACACTGAGGTTACctcgagGTAACCTCAGTGTGTTAGTCC
Adenoviral shRNA plasmids were generated in Cyagen using Vectorbuilder. The following sequences were inserted into pAV[shRNA]_eGFP-U6 vector.
Ad-sh_Negctl:
CAACAAGATGAAGAGCACCAAcgaaTTGGTGCTCTTCATCTTGTTG
Ad_sh_*lnc-Prdm16*:
GCAGCTTGATTACTTACAAGAcgaaTCTTGTAAGTAATCAAGCTGC

**Retroviral infection and siRNA transfection**. Retrovirus were produced by co-transfection of retroviral plasmids and packing plasmid pCL-Eco into 293T cells from American Type Culture Collection. Culture medium was changed to fresh medium at 16–18 h after transfection and viruses were collected at 48 h after transfection. Primary pre-adipocytes were infected with fresh viruses at ~60–70% confluence and then cultured to confluence, followed by standard differentiation protocol for 5 days.

For siRNA transfection, pre-adipocytes were grown to 90–95% confluence and transfected with siRNAs (100 nM) with Lipofectamine® RNAiMAX according to the manufacturer's instruction. Twelve hours after transfection, media were replaced and cells were grown to confluence for differentiation induction. Four days after induction, adipocytes were harvested for downstream analysis. siRNA sequences are listed in Supplementary Table 2.

**Adenovirus injection**. Adenovirus was generated according to the standard protocol and purified with cesium chloride gradient centrifugation[46]. We anesthetized 7-week-old C57Bl/6 mice, incised a small opening on the skin of the lateral region to expose the iWAT, and locally injected 30 μl of adenoviral shRNAs at $1 \times 10^{12}$ vp/ml into iWAT. The skin incision was closed with sutures. Seven days after injection, mice were exposed at 4 °C for 24 h to induce iWAT browning before tissue harvest. We conducted similar procedure to inject virus into interscapular brown fat and harvested the BAT 7 days after injection. We did not expose animals with interscapular injection to cold exposure, because cold temperature will cause animals to hunch and the surgical wounds will burst open.

**cDNA synthesis and qPCR**. Total RNA was isolated from cultured cells or tissues using miRNeasy kit (Qiagen) and reverse transcribed using random primers (M-MLV Reverse Transcriptase, Promega). Gene-specific primers were used for PCR amplification, followed by Sybr Green-based qPCR performed in Applied Biosystems 7900HT Fast Real-time PCR System. RPL23 was used as an internal control for normalization. Primer sequences are listed in Supplementary Table 3.

**Western blot**. Western blot was performed with Anti-Pgc1α (sc-13067, 1:1000 dilution) from Santa Cruz Biotech, Anti-Ucp1 from Abcam (ab23841, 1:2000 dilution), and β-actin from Sigma (A1978, 1:3000 dilution).

**Oil Red O staining**. Oil Red O (ORO) solution was prepared by dissolving 0.15 g ORO in 30 ml isopropanol, mixed with 20 ml water, and filtered through filter paper. Cultured brown and white mouse adipocytes at differentiation day 5 were washed twice with PBS, fixed with 10% formalin at room temperature for 5 min, washed off, and incubated for at least 1 h with fresh formalin. Formalin-fixed cells were then stained with ORO solution for 60 min. Finally, stained cells were washed with $H_2O$ to remove ORO residues.

**RNA-seq transcriptome assembly and lncRNA identification**. Total RNA was extracted from human fetal interscapular BAT, adult oWAT, and adult sWAT using a QIAGEN kit. Strand-specific poly(A) RNA-seq libraries were prepared according to NEBNext Ultra Directional RNA Library Prep Kit for Illumina and sequenced on Hiseq2000 sequencer. To de novo reconstruct lncRNA catalog, we closely followed the steps used in previous publication[24], with the exception of replacing mouse with human reference genome and annotations. Briefly, the 100 bp paired-end reads were aligned to Hg19 using Tophat v 2.0.11[47] and de novo-assembled using Cufflinks v 2.2.1[34]. To identify reliable, multi-exonic long non-coding transcripts, we implemented the following selection criteria: (i) read coverage ≥3 in at least one of the tissues; (ii) ≥200 bp; (iii) ≥2 exons; (iv) low predicted protein-coding potential; and (v) no sense overlap with known coding genes derived from UCSC, Ensembl, and Resfeq databases. All generated sequencing data that support the findings of this study have been deposited in Gene Expression Omnibus (GEO) database with the accession number GSE97205.

**RNA-seq and analysis**. Total RNAs were extracted with Qiagen miRNeasy kit. Strand-specific RNA-seq libraries were prepared using NEBNext Ultra Directional RNA Library Prep Kit for Illumina and sequenced on Hiseq2000 sequencer platform. The 100 bp paired-end reads were aligned against Hg19 or mm10 using Tophat (version-2.0.11)[48]. Gene expression was subsequently quantified using Cufflinks (version 2.1.1) into units known as FPKM[48]. Within each pairwise comparison, genes with low expression (mRNA FPKM ≤ 1 or lncRNA FPKM ≤ 0.1 in both samples) were excluded from all downstream analyses.

**Conversed lncRNA identification**. For each human lncRNA, we extended ±500 kb and examined all protein-coding genes found within this window size on the same strand. Next, we made use of previously published mouse adipose-expressed lncRNAs[24] and extracted protein-coding genes found within this window size (±500 kb) on the same strand. A human lncRNA was considered to have a mouse ortholog if the relative order of protein-coding genes with mouse lncRNA is preserved.

**Tissue specificity analysis of mRNA and lncRNA**. We downloaded Hg19-aligned RNA-seq reads from Human BodyMap[33] and used Cufflinks to quantify lncRNA and mRNA expression levels based on de novo-constructed lncRNA models and Ensembl annotation, respectively. Tissue specificity score for each gene in a given tissue (lncRNA or mRNA) was given as the proportion of its expression against the cumulative expressions of this gene across all 22 analyzed tissues. For each gene, we ranked the specificity scores across all tissues and defined the highest score as maximal fractional expression. To identify tissue-specific genes, we used a minimal threshold of 0.1 after benchmarking the specificity scores against known tissue-specific genes such as *Ucp1*. By this, we assigned a gene to be specifically expressed in a given tissue if it shows maximal specificity score in this tissue, which is also above a threshold of 0.1.

**Gene set enrichment analysis**. We performed GSEAs on two pre-ranked gene lists obtained from human BAT and WAT adipogenesis cell cultures using GSEA[37]. Genes were pre-ranked by $\log_2$ expression change at differentiation day 21 relative to day 0, subsequently compared against "GO gene sets" (C5) and "Hallmarks" in MSigDB using default parameters.

**Gene Ontology analysis**. Selected gene lists were analyzed using DAVID[49] to identify enriched Gene Ontology (GO). We only considered significant ($p < 0.05$) Biological Process (GOTERM_BP_FAT) and Molecular Function (GOTERM_MF_FAT), which contained at least three genes in each GO term. Canonical pathways and network analysis were performed with Ingenuity Pathway Analysis (Build version 400896M).

**Data availability**. All generated sequencing data that support the findings of this study have been deposited in GEO database with the accession number GSE97205. All other analyzed sequencing data for human clonal cell lines[19] and cold exposed human adipose[38] are available in ArrayExpress, with accession numbers E-MTAB-2602 and E-MTAB-4031, respectively.

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

## Acknowledgements

RNA samples from human adipose tissue are the generous gifts from Zen-Bio Inc. This work was supported by Singapore NRF fellowship (NRF-2011NRF-NRFF 001-025), Tanoto Initiative in Diabetes Research to L.S., National Medical Research Council's Cooperative Basic Research Grant (CBRG; NMRC/CBRG/0070/2014 and NMRC/CBRG/0101/2016), Open Fund-Individual Research (OF-IRG) Grant (NMRC/OFIRG/0062/2017), and Ministry of Education (MOE) Tier2 grant (MOE2017-T2-2-009). This work was supported by the RNA Biology Center at CSI Singapore, NUS, from funding by the Singapore Ministry of Education's Tier 3 grants, grant number MOE2014-T3-1-006. This work was also supported by the Recruitment Program for Young Professionals (C.D.), Zhejiang Key Subject of Medical Science (C.D.), National Natural Science Foundation of China (81700770), and Zhejiang Provincial Natural Science Foundation of China (LY18C060006).

## Author contributions

D.X., C.D., L.S., Q.C., and M.K.-S.L. conceived and designed the experiment. D.X., S.Y.C., S.X., A.C.E.W., Y.Z., D.Z., Z.S., and K.A.L. performed the experiments. Y.C.L. and C.D. performed the bioinformatics analysis. C.D., L.S., Y.C.L., and D.X. wrote the manuscript.

## Additional information

**Competing interests:** The authors declare no competing interests.

