## [Peer Review File · Nature Communications]

Reviewers' comments:

Reviewer #1

Remarks to the Author:

In this study, Lim et al. strive to (re)-construct a novel and useful roadmap for human long noncoding RNAs based on deep RNA-Sequencing data from fetal adipose depots. Computational models were applied to demonstrate that lncRNAs are indeed better tissue 'markers' (no causal relationship with BAT function can be assumed at this step) than coding mRNA using their own, but also corroboration with public expression data. This catalogue of human (conserved) lncRNAs is useful for the community and the computational approach (which has been published and described in detail by the same group for mouse adipose tissue; Alvarez-Dominguez_2015_Cell Metab) can be applied to other tissues / organisms. Hereafter, Lim et al. focus on a particular divergent lncRNA opposite to PRDM16 (lnc-dPRDM16), a key (coding) regulator of BAT identity and, using RNAi in primary cells, confirm that lnc-dPRDM16 positively controls brown adipogenesis in cell culture.

Collective, the dataset of human adipose depot-specific lncRNAs is novel and the computational models applied solid. This compendium of hs lncRNAs will be instrumental for the community, yet the novelty of transferring published transcriptome assembly approaches from mouse to human is limited. The finding that lnc-dPRDM16 is functionally relevant for brown adipogenesis needs to be supported with more in vivo data and the contribution of lnc-dPRDM16 to metabolism, energy expenditure and thermogenesis should be addressed given the pivotal role of BAT in these processes.

Minor suggestions:

- A (UCSC / ENSEMBL) gene browser screenshot of lnc-dPRDM16 and an estimate of its abundance, subcellular localisation, coding potential etc. should be given
- To rule out that lnc-dPRDM16 is broadly regulating adipogenesis the same RNAi experiments should be performed in VAT / SCAT cell systems.
- Relevant information pertaining to public RNA-Seq (eg E-MTAB-2602) should be given: Relevant QC measures like mappability of the dataset, PCAs, etc

Reviewer #2

Remarks to the Author:

The manuscript "De novo reconstruction of human adipose reveals conserved lncRNAs as regulators of brown adipogenesis" describes the generation of RNAseq data, reconstruction of transcripts and identification of novel long non-coding RNAs in three human adipose tissues. The methodologies used are standard the the filtering criteria applied are reasonable. Introduction is clearly written and appropriately referenced and the remainder of the paper is also clearly written for the most part.

The authors establish that the lncRNAs are novel by comparison with other reference gene annotation, tissue-specific and dynamic in their expression.

The authors also ask whether lncRNAs exert a cis regulatory effect on genes related to brown adipogenesis. While they establish correlation between lncRNAs and nearby protein-coding genes, however, there is no evidence presented to confirm that the lncRNAs are regulating the expression of coding genes.

The authors identify a lncRNA (lnc-dPRDM16 in the manuscript), divergently expressed to a protein-coding gene, the adipogenesis master regulator PRDM16. The lncRNA and coding gene

show strongly correlated expression in both human and mouse which is suggested to support a role for lnc-dPRDM16 in adipocyte biology. However, there is no evidence presented to confirm their correlated expression reflects anything more than regulation via common regulatory elements.

The authors also describe knocking down lnc-dPRDM16, which has the effect of downregulating a divergently expressed protein-coding gene, the adipogenesis master regulator PRDM16. Other markers of adipogenesis are also subsequently downregulated. Again, it is suggested that this indicates a functional role for the lncRNA. However, it is possible that the shRNA method used to knockdown lnc-dPRDM16 is not selectively suppressing the lncRNA but rather triggering RNA-directed transcriptional silencing via DNA methylation and histone modifications of promoter elements shared with PRDM16. The authors must exclude this as a possibility before being confident in assigning function to lnc-dPRDM16.

minor comments

“maximal fractional expressionof at least 0.1 as the threshold” - requires further explanation

“depot-specific” - should be tissue-specific

Reviewers' comments:

Reviewer #1 (expert in adipose tissue biology and regulatory RNAs)
Remarks to the Author:

In this study, Lim et al. strive to (re)-construct a novel and useful roadmap for human long noncoding RNAs based on deep RNA-Sequencing data from fetal adipose depots. Computational models were applied to demonstrate that lncRNAs are indeed better tissue 'markers' (no causal relationship with BAT function can be assumed at this step) than coding mRNA using their own, but also corroboration with public expression data. This catalogue of human (conserved) lncRNAs is useful for the community and the computational approach (which has been published and described in detail by the same group for mouse adipose tissue; Alvarez-Dominguez_2015_Cell Metab) can be applied to other tissues / organisms. Hereafter, Lim et al. focus on a particular divergent lncRNA opposite to PRDM16 (lnc-dPRDM16), a key (coding) regulator of BAT identity and, using RNAi in primary cells, confirm that lnc-dPRDM16 positively controls brown adipogenesis in cell culture.

Collective, the dataset of human adipose depot-specific lncRNAs is novel and the computational models applied solid. This compendium of hs lncRNAs will be instrumental for the community, yet the novelty of transferring published transcriptome assembly approaches from mouse to human is limited. The finding that lnc-dPRDM16 is functionally relevant for brown adipogenesis needs to be supported with more *in vivo* data and the contribution of lnc-dPRDM16 to metabolism, energy expenditure and thermogenesis should be addressed given the pivotal role of BAT in these processes.

We agree that *in vivo* evidence is important to establish the role of lnc-dPRDM16. We generated an adenoviral shRNA and locally injected it into BAT (Figure S9) and inguinal WAT (Figure 9). The shRNA successfully knocked down lnc-dPRDM16 *in vivo in BAT* (Figure S9) and during WAT browning (Figure 9), which resulted in impaired BAT-selective gene expression (Figure 9, Figure S9). Thus, lnc-dPRDM16 is required for supporting BAT program *in vivo*.

We acknowledge that this approach will not allow us to assess the impact of lnc-dPRDM16 knockdown on systemic metabolism because the knockdown is limited to a specific depot for only a short period (1-2 weeks). However, the goal of this manuscript is to generate and characterize a comprehensive human lncRNA catalog for adipose tissue, and provided a roadmap to identify functionally important lncRNAs in adipose tissue. Our presented data have demonstrated that lnc-dPRDM16 is an important regulator for BAT and iWAT. We believe that a further comprehensive metabolic phenotyping is beyond the scope of this manuscript.

Minor suggestions:

- A (UCSC / ENSEMBL) gene browser screenshot of lnc-dPRDM16 and an estimate of its abundance, subcellular localisation, coding potential etc. should be given

Thanks for your suggestions. We have included this information about lnc-dPRDM16 in our manuscript. A (UCSC / ENSEMBL) gene browser screenshot of lnc-dPRDM16 has been included in Figure 7A. It is expressed at 0.967 FPKM in human BAT and at 5.26 FPKM in mouse BAT. Using phyloCSF, its predicted coding potential was -190.75. We also examined its localization in brown adipose tissue and found majority of this lncRNA is in cytosol (FigureS7).

-To rule out that lnc-dPRDM16 is broadly regulating adipogenesis the same RNAi experiments should be performed in VAT / SCAT cell systems.

We agree that it is important to address whether this lncRNA may affect general adipogenesis and/or BAT-selective features. In our point of view, these two aspects are not mutually exclusive --- a gene can be essential during general adipogenesis and also required for BAT-feature maintenance after differentiation. We have performed a few experiments to address this question.

1) As suggested, we knocked down this lncRNA in primary subcutaneous adipocyte culture (Figure S8). Ablation of lnc-dPRDM16 also impaired subcutaneous adipocyte differentiation, so this lncRNA does play a role during adipogenesis, which is consistent with its upregulation during brown and white adipogenesis (Figure 8C).

2) To test whether lnc-dPRDM16 contribute to adipocyte feature maintenance in mature brown adipocytes, we generated adenoviral shRNAs and locally injected into BAT in mice which is mainly composed of mature brown adipocytes. Knockdown lnc-dPRDM16 *in vivo* did impair BAT-selective markers (Figure S9) without affecting pan-adipogenic markers (Figure S9).

3) In inguinal adipose tissue, knockdown of this lncRNA affect both pan-adipogenesis markers and BAT-selective markers during browning. (Figure 9).

Taken together, we demonstrated that lnc-dPRDM16 has a pivotal role in adipocytes. It is required for general adipogenesis. In mature adipocytes, it is needed for maintaining a full adipogenic program in iWAT and for BAT-selective program in BAT.

-Relevant information pertaining to public RNA-Seq (eg E-MTAB-2602) should be given: Relevant QC measures like mappability of the dataset, PCAs, etc

As suggested, we have included the related information in Table S1, Figure S3D and Figure S3E in the revised manuscript.

Reviewer #2 (expert in transcriptomics)
Remarks to the Author:

The manuscript "De novo reconstruction of human adipose reveals conserved lncRNAs as regulators of brown adipogenesis" describes the generation of RNAseq data, reconstruction of transcripts and identification of novel long non-coding RNAs in three human adipose tissues. The methodologies used are standard the filtering criteria applied are reasonable. Introduction is clearly written and appropriately referenced and the remainder of the paper is also clearly written for the most part.

The authors establish that the lncRNAs are novel by comparison with other reference gene annotation, tissue-specific and dynamic in their expression.

The authors also ask whether lncRNAs exert a cis regulatory effect on genes related to brown adipogenesis. While they establish correlation between lncRNAs and nearby protein-coding genes, however, there is no evidence presented to confirm that the lncRNAs are regulating the expression of coding genes.

Thanks for this suggestion. We agree that this point in our manuscript should be strengthened. Accordingly, we have knocked down 6 additional lncRNAs and examined the expression of their nearest protein genes. For 5 out of 6 lncRNAs, knockdown results in decreased expression of their nearby genes. This data strongly suggests that the conserved correlation between lncRNA-mRNA expression in our study (Figure 6) is not just limited to expression association but does imply functional regulation. This is consistent with an earlier study about the regulatory role of divergent lncRNAs on nearby gene expression¹. Furthermore, we also point out in the text that the multiple mechanism may underlie these regulations. The influence of these cis elements may or may not rely on the sequences of these lncRNAs themselves. It is possible that the general process of these transcripts such as splicing may mediate the crosstalk, which was observed in other studies³.

The authors identify a lncRNA (lnc-dPRDM16 in the manuscript), divergently expressed to a protein-coding gene, the adipogenesis master regulator PRDM16. The lncRNA and coding gene show strongly correlated expression in both human and mouse which is suggested to support a role for lnc-dPRDM16 in adipocyte biology. However, there is no evidence presented to confirm their correlated expression reflects anything more than regulation via common regulatory elements.

The reviewer concerns that the correlated expression could be merely via common regulatory element (their shared promoter). In our study, when we knocked down lnc-dPRDM16, we found a clear decrease of prdm16 transcripts (Figure 8H), which suggested a regulatory function of lnc-dPRDM16 on prdm16 expression. We acknowledged that the decreased expression of prdm16 might be due to an inhibition of general adipogenesis because we knocked down this lncRNA during brown adipocyte differentiation in cell culture. To address this issue, we generated adenoviral shRNAs and locally injected them into BAT and inguinal WAT where majority of cells are mature adipocytes and the influence on adipogenesis is minimal during the experimental duration. When we knocked down lnc-dPRDM16 in iWAT, the expression of PRDM16 was indeed down regulated. Since adipogenesis is unlikely to be affected during the experiment duration (7 days), this data strongly supports that lnc-dPRDM16 can exert regulatory influence on the divergent mRNA. Moreover, this is consistent with recent study that the divergent lncRNAs tend to regulate the expression of the nearby gene³.

We agree with the reviewer that lnc-dPRDM16 can also function independently from Prdm16. When we knocked down lnc-dPRDM16 in BAT (Figure S9), we observed a decreased expression of BAT-markers but not PRDM16. Thus, the regulatory influence of lnc-dPRDM16 on Prdm16 is depot-specific. We have emphasized this point in the text.

Importantly, we would like to emphasize that the main goal of this manuscript is to generate and characterize a comprehensive human lncRNA catalog for adipose tissue (brown adipose tissue in particular), and to provide a roadmap to identify functionally important lncRNAs. We believe that this goal has been achieved by the presented data. The mechanisms used by lnc-dPRDM16 to regulate adipogenesis and maintain adipocyte program remain to be further characterized in a future study.

The authors also describe knocking down lnc-dPRDM16, which has the effect of downregulating a divergently expressed protein-coding gene, the adipogenesis master regulator PRDM16. Other markers of adipogenesis are also subsequently downregulated. Again, it is suggested that this indicates a functional role for the lncRNA. However, it is possible that the shRNA method used to knockdown lnc-

dPRDM16 is not selectively suppressing the lncRNA but rather triggering RNA-directed transcriptional silencing via DNA methylation and histone modifications of promoter elements shared with PRDM16. The authors must exclude this as a possibility before being confident in assigning function to lnc-dPRDM16.

Concern is raised regarding whether the phenotypes observed were due to silencing the PRDM16 promoter which is independent of the siRNA-mediated posttranscriptional cleavage of mRNA. The phenotype observed in our study is very unlikely due to the promoter silencing mechanism. To generate siRNA-mediated promoter silencing, the siRNAs need to be designed to directly target the promoter region^{4,5}. Morris *et al.* demonstrated that siRNAs targeting the gene body could not induce transcriptional silencing⁵. In our study, both shRNAs were designed to target the 2nd exon of lnc-dPRDM16 which was more than 5kb away from the prdm16 promoter region. It is unlikely to induce promoter silencing from such a distance.

To further address the reviewer's concern, we performed shRNA knockdown during brown adipogenesis in the presence of 5-azacytidine (5-azaC), an inhibitor of DNA methyltransferases as we did in our earlier work⁶. Inhibiting DNA methylation doesn't affect the phenotype of lnc-dPRDM16 knockdown as shown below, which further supports that our observation was not due to promoter silencing.

minor comments

"maximal fractional expressionof at least 0.1 as the threshold" - requires further explanation

Knockdown of lnc-dPRDM16 during brown adipogenesis in the presence of 5-AZA. Brown preadipocytes were infected by retroviral shRNAs targeting lnc-dPRDM16. Cells were treated with 5-azacytidine (10uM) for 48 hours before differentiation. RNAs were harvested at day 4 of differentiation for real-time PCR.

Thanks for pointing this out. We have added elaboration to Experimental procedure, section "Tissue specificity analysis of mRNA and lncRNA".

"depot-specific" - should be tissue-specific

We have made the changes as suggested.

References

1. Luo S, *et al.* Divergent lncRNAs Regulate Gene Expression and Lineage Differentiation in Pluripotent Cells. *Cell stem cell* **18**, 637-652 (2016).
2. Chondronikola M, *et al.* Brown Adipose Tissue Activation Is Linked to Distinct Systemic Effects on Lipid Metabolism in Humans. *Cell metabolism* **23**, 1200-1206 (2016).
3. Engreitz JM, *et al.* Local regulation of gene expression by lncRNA promoters, transcription and splicing. *Nature* **539**, 452-455 (2016).
4. Hawkins PG, Santoso S, Adams C, Anest V, Morris KV. Promoter targeted small RNAs induce long-term transcriptional gene silencing in human cells. *Nucleic Acids Res* **37**, 2984-2995 (2009).
5. Morris KV, Chan SW, Jacobsen SE, Looney DJ. Small interfering RNA-induced transcriptional gene silencing in human cells. *Science* **305**, 1289-1292 (2004).
6. Lim YC, Chia SY, Jin S, Han W, Ding C, Sun L. Dynamic DNA methylation landscape defines brown and white cell specificity during adipogenesis. *Molecular metabolism* **5**, 1033-1041 (2016).

REVIEWERS' COMMENTS:

Reviewer #1 (Remarks to the Author):

Major reviewers' critiques were sufficiently addressed. Although the authors' themselves admit that deeper insights into the molecular mechanism of how lnc-PRDM16 affects Prdm16 transcript levels are lacking (triple helix formation, changes in histone landscapes around common regulatory elements, higher-order chromosome conformation are only a few potential explanations), these experiments fall into the scope of an independent manuscript.

Reviewer #2 (Remarks to the Author):

The authors have done considerable additional work to address the initial concerns of the reviewers. The additional experimental work supports the major assertions made in the paper. However, I would still suggest that in the paragraph starting "To ask if lncRNA may exert a cis regulatory effect on genes related to brown adipogenesis", while correlation is identified, no evidence for causation is presented (at this point).

Reviewer #1 (Remarks to the Author):

Major reviewers' critiques were sufficiently addressed. Although the authors' themselves admit that deeper insights into the molecular mechanism of how lnc-PRDM16 affects Prdm16 transcript levels are lacking (triple helix formation, changes in histone landscapes around common regulatory elements, higher-order chromosome conformation are only a few potential explanations), these experiments fall into the scope of an independent manuscript.

Thanks for the positive comments from this reviewer.

Reviewer #2 (Remarks to the Author):

The authors have done considerable additional work to address the initial concerns of the reviewers. The additional experimental work supports the major assertions made in the paper. However, I would still suggest that in the paragraph starting "To ask if lncRNA may exert a cis regulatory effect on genes related to brown adipogenesis", while correlation is identified, no evidence for causation is presented (at this point).

We agreed with the reviewers that although our data, including our knockdown data in last version (Fig. 6C-H), demonstrate that there are regulatory interactions between lncRNAs and its nearby mRNAs, we can't 100% preclude the possibility that such interactions might be indirect. We have edited several places in the text to tone down our statement by pointing out alternative possibilities. For example,

"We observed as many as 599 out of 711 (84.2%) of the lncRNA-mRNA pairs being positively correlated (Fig. 4C), suggesting that the paired lncRNAs and mRNAs may regulate the transcription of each other positively *in cis* or they may share common regulatory elements."

"It is also possible that some lncRNAs do not affect their nearby mRNAs directly but through secondary effects from altered cellular status."

We believe that these changes make our statements less biased and more accurately.